# Nitrosative Stress and Human Disease: Therapeutic Potential of Denitrosylation

**DOI:** 10.3390/ijms22189794

**Published:** 2021-09-10

**Authors:** Somy Yoon, Gwang Hyeon Eom, Gaeun Kang

**Affiliations:** 1Department of Pharmacology, Chonnam National University Medical School, Hwasun 58128, Korea; oouate@naver.com; 2Division of Clinical Pharmacology, Chonnam National University Hospital, Gwangju 61469, Korea

**Keywords:** nitrosation, S-nitrosylation, human disease, denitrosylation, therapeutics

## Abstract

Proteins dynamically contribute towards maintaining cellular homeostasis. Posttranslational modification regulates the function of target proteins through their immediate activation, sudden inhibition, or permanent degradation. Among numerous protein modifications, protein nitrosation and its functional relevance have emerged. Nitrosation generally initiates nitric oxide (NO) production in association with NO synthase. NO is conjugated to free thiol in the cysteine side chain (S-nitrosylation) and is propagated via the transnitrosylation mechanism. S-nitrosylation is a signaling pathway frequently involved in physiologic regulation. NO forms peroxynitrite in excessive oxidation conditions and induces tyrosine nitration, which is quite stable and is considered irreversible. Two main reducing systems are attributed to denitrosylation: glutathione and thioredoxin (TRX). Glutathione captures NO from S-nitrosylated protein and forms S-nitrosoglutathione (GSNO). The intracellular reducing system catalyzes GSNO into GSH again. TRX can remove NO-like glutathione and break down the disulfide bridge. Although NO is usually beneficial in the basal context, cumulative stress from chronic inflammation or oxidative insult produces a large amount of NO, which induces atypical protein nitrosation. Herein, we (1) provide a brief introduction to the nitrosation and denitrosylation processes, (2) discuss nitrosation-associated human diseases, and (3) discuss a possible denitrosylation strategy and its therapeutic applications.

## 1. Introduction

### 1.1. Posttranslational Modification

Posttranslational modification of protein refers to dynamic modifications after the termination of translation. Generally, posttranslational modification involves the ligation or removal of a small chemical moiety from the side chain of an amino acid through acetylation, methylation, phosphorylation, etc. Small-sized proteins, such as ubiquitin, interferon-stimulated gene 15, and small ubiquitin-related modifier, can also be added [1]. Posttranslational modification enables immediate changes to the function of target proteins. For example, the agonist-bound epidermal growth factor receptor induces dimerization and autophosphorylation, which elicits the recruitment of a downstream signaling molecule, including phosphoinositide 3-kinase, Ras GTPase, or signal transducer, and is an activator of the transcription protein. The phosphorylation-dependent signal cascade initiates gene transcription or cell cycle progression, which promote the growth of various solid tumors, including epidermal cell carcinoma [2]. On the other hand, phosphorylation can negatively regulate enzymatic function. Glycogen synthase kinase (GSK) 3 is a well-known kinase whose intrinsic activity is differentially regulated by phosphorylation. GSK3 exists in two isoforms—GSK3α and GSK3β—and phosphorylates more than 100 substrates in various disease progressions as a kinase. Phosphorylation at tyrosine 279 in GSK3α or at tyrosine 216 in GSK3β activates the kinase function, while phosphorylation at serine 21 in GSK3α or at serine 9 in GSK3β inactivates the kinase function. Phosphorylation can modulate the kinase activity of GSK3 in either direction depending on the modification site [3].

Besides phosphorylation, methylation is one of the most widely studied posttranslational modifications in terms of major histone modification. Methylation takes place in lysine or arginine. Typically, mono- or di-methylation is observed in arginine, but mono-, di-, or tri-methylation are all detected in lysine [4,5]. For a long time, di-methylation and tri-methylation were considered to be involved in transcription repression by heterochromatin formation. Tri-methylation in histone 3 lysine 9 (H3K9) induces the compaction of chromatin and the localization of heterochromatin in the central region. However, di-methylation in H3K9 forces it to move to the nuclear periphery and allows its interaction with lamin B [6]. Both the localization and transcription of lamina-associated domains among the di-methylation area are specifically and dynamically regulated by interactions with the lamina protein. Although the methylation in H3K9 represents a di- or tri-methylation, histone methylation generally results in transcription silencing [7]. However, methylation in H3K4 is closely associated with transcription activation. Although no definite evidence has been provided, it is generally accepted that the nucleosome remodeling factor recognizes H3K4 methylation and maintains the euchromatin status of the methylated regions [8].

### 1.2. Nitric Oxide Modification

Classical posttranslational modifications, such as phosphorylation, acetylation, methylation, and ubiquitination, have been extensively studied for a long time. However, protein nitrosation is a recently highlighted protein modification. Nitrosation is a general term that indicates nitrogen-associated gaseous modification and includes the ligation of nitric oxide (NO) or nitrogen dioxide on proteins, peptides, and metal ions [9].

NO is synthesized by NO synthase (NOS) during the catalysis of L-arginine into L-citrulline. Three NOS isoforms have been reported: neuronal NOS (nNOS and NOS1), inducible NOS (iNOS and NOS2), and endothelial NOS (eNOS and NOS3). nNOS is highly expressed in neuronal cells, including those in the brain and skeletal and in cardiac muscle. nNOS is predominantly involved in neurotransmission through N-methyl-D-aspartate (NMDA) receptor coupling [10]. In response to cytokine stimulation, mononuclear leukocytes produce iNOS and subsequently differentiate into a special subtype of macrophages. iNOS is also closely linked to the regulation of vascular permeability and to the migration of immune cells. eNOS is a major subtype observed in endothelial cells and mainly regulates the vascular tone for smooth muscle relaxation. eNOS is constitutively expressed but it can be aberrantly activated by shear stress and can deteriorate vascular integrity [11].

Besides NOS subtypes, NO can be supplied from other sources. Iron ions in the center of the hemoglobin deliver oxygen (O_2_) from the lungs to the peripheral organs. Like O_2_, NO is also bound to the beta chain in globin or is directly conjugated to iron [12,13]. When circulating red blood cells sense hypoxic environments, NO is released from hemoglobin. Free NO gas simply diffuses into vascular smooth muscle cells and induces relaxation, which finally results in an increase in the size of the vascular lumen [13]. Just as the human body utilizes red blood cells as a transporter for the long-distance transport of NO, intracellular molecules tightly regulate NO homeostasis. Glutathione (GSH) is one of the most abundant free thiol groups in human cells [14]. To remove reactive O_2_ species (ROS), GSH undergoes oxidization and forms the homodimer GSH disulfide (GSSG). The oxidized state of GSH, GSSG, can be converted to the reduced state, GSH, in the presence of NAPDH. Similarly, free thiol in GSH is utilized as a reservoir of NO in the cell. GSH captures intracellular NO originally generated by NOSx and forms S-nitrosoglutathione (GSNO). GSNO serves as a stable pool of bioavailable NO. GSNO reductase (GSNOR; alternative name formaldehyde dehydrogenase, a class III alcohol dehydrogenase) can metabolize GSNO to GSH sulfonamide, which finally removes the NO moiety in the cell. Hence, the intracellular GSNO level can be an indirect parameter for the estimation of the endogenous NO level [15].

The most well-known direct function of NO is vasodilation, which is secondarily induced by the activation of soluble guanylyl cyclase (sGC). sGC is a heme-protein that functions as an intracellular receptor for NO. When free NO binds to sGC, it leads to an enzymatic activation of over 300-fold compared with that in the unbound state [16]. In the presence of NO, Fe(II)heme in sGC catalyzes GTP into cGMP, which directly or indirectly activates its substrates, such as the cation channels, protein kinase G (PKG), protein kinase B, and myosin phosphatase for muscle relaxation. The heme of sGC forms a weak bond between iron and histidine, which plays an important role in acquiring the selective binding ability of NO over O_2_. Because the sGC heme binding to NO has been shown to be greater than that of any other known hemoprotein, sGC might act as NO scavengers in cells [17].

It is noteworthy that NO generates the secondary messenger cGMP in cells after the activation of sGC, which is catalyzed by the phosphodiesterase (PDE) superfamily [18]. PDE4/7/8 selectively degrades the cyclic adenosine monophosphate (cAMP), whereas PDE5/6/9 catalyzes cGMP. The other subtypes of PDE, PDE1/2/3/10/11, hydrolyze both cAMP and cGMP [19]. When the PDE activity is suppressed, the half-life of cAMP or cGMP is remarkably increased, which potentiates the biological function of NO. Indeed, subtype-specific PDE inhibitors, such as the PDE5 inhibitor, can improve blood supply via relaxation of the vasculature [20]. It has been reported that several subtypes of PDE may be involved in chronic human diseases, such as neurodegenerative disease [21], cardiovascular disease [22], pulmonary disease [23], chronic inflammatory disease [24], and others [25].

Interestingly, cGMP−cAMP cross talk is well established in the heart. NO can activate or inactivate the cAMP system in a PDE subtype-dependent manner. PDE2 or PDE3 non-selectively degrade cAMP and cGMP at similar rates [26]. cGMP potentiates the enzyme activity of PDE2 through its binding, which in turn accelerates the degradation of cAMP [27]. On the contrary, PDE3 predominantly hydrolyzes cAMP with a higher binding affinity to cGMP. The enzymatic activity of PDE3 is inhibited in the presence of cGMP, which in turn augments the cAMP half-life [28]. The NO/cGMP axis regulates the vascular tone, while cAMP/PKA signal transduction governs cardiac contractility. Hence, this cGMP−cAMP cross talk is a notable therapeutic target for the dual goals of a positive inotropic effect with a negative preload or afterload property.

Although NO predominantly produces cGMP as a secondary messenger, it stimulates protein nitrosylation in a cGMP-independent manner [29]. However, it is unclear whether the derangement of the NOSx activity plays a role in PDE-associated human diseases; further studies are required on this topic.

### 1.3. S-Nitrosylation

Oxidative stress stimulates the production of ROS, including superoxide O_2_^−^ and peroxynitrite (ONOO^−^), a highly reactive free radical. These damaging free radicals oxidize several lipoproteins and amino acids in many proteins. Nitration can occur in several amino acids, such as cysteine, methionine, tryptophan, and tyrosine, but generally tyrosine nitration is predominantly observed. Nitrosation of tyrosine forms 3-nitrotyrosine, which is stable, and this is regarded as an irreversible modification [30]. No enzymes have yet been discovered that remove nitrous acid from the aromatic ring. Therefore, tyrosine nitration is a useful biomarker to measure the accumulated intracellular oxidative stress [31].

Another typical example of the nitrosation of a protein is the S-nitrosylation of cysteine. The S in S-nitrosylation indicates the presence of a sulfur atom in the cysteine amino acid, and nitrosylation means the attachment of NO. NO covalently attaches to the thiol group (-SH) of cysteine, and finally, participates in dynamic modifications, such as metal ion conjugation, disulfide bridge formation, and nitrosylation. Although S-nitrosylation is a gaseous modification, sequential transnitrosylation by direct physical binding is more widely accepted rather than the attachment of free NO [32]. To detect the S-nitrosylation level, the NO moiety has to be converted to a stable conjugated state because nitrosylated products are easily reduced, even by high temperatures or ultraviolet wavelengths. Experimentally, the biotin switching assay is widely used to visualize S-nitrosylation [33].

Recently, many reports have suggested that S-nitrosylation exists in the temporal state to form permanent modifications, depending on the redox level [34,35,36]. S-nitrosylation progresses to sulfenic acid, to disulfide (SS), to sulfinic acid (SO_2_H), and to irreversible sulfonic acid in response to nitrosative or oxidative stresses. Previous studies have pointed out that S-nitrosylation possibly exists as the beginning of thiol-modification, but they also agree that S-nitrosylation is mainly involved in the signal transduction and reversible regulation of the protein function [37].

## 2. Human Disease and Nitrosation

### 2.1. Central Nervous System

Among the three subtypes of NOS, nNOS is predominantly expressed in the general neurons. In the physiologic condition, nNOS exists in the soluble form and composites the functional compartment in association with the postsynaptic density of 95 complexes that are anchored to the membrane NMDA receptor. The NMDA receptor opens its channel pore in response to glutamate binding and transits calcium ions into the cell. Soluble nNOS is activated in the presence of calcium ions and produces NO, which, in turn, stimulates sGC or transnitrosylation cascades. Interestingly, the NMDA receptor activity can be finely modulated by S-nitrosylation. The NMDA receptor is a heterotetramer composed of subunits GluN1, GluN2, and GluN3 [38]. One NMDA receptor subunit, GluN2B, serves as a target molecule of S-nitrosylation. S-nitrosylation at Cys 399 changes the conformational structure, which, in turn, enhances the agonist binding affinity to the receptor. Initial S-nitrosylation promotes the sensitization of the agonist, but it ultimately causes receptor desensitization. S-nitrosylation of GluN2B participates in the transient calcium homeostasis loop in response to the sustained calcium influx [39].

#### 2.1.1. Neurogenesis and S-Nitrosylation

Limited evidence supports that nNOS is involved in neurogenesis. Two special zones in the dentate gyrus—the subgranular and subventricular zones—are well-established neurogenesis focuses in the adult brain. Chronic treatment of N(ω)-nitro-L-arginine methyl ester (L-NAME), a non-selective NOS inhibitor (NOSi), or 7-nitroindazole, a relatively specific nNOSi, in adult mice significantly improved BrdU incorporation in the subventricular zone without any alteration of neuronal migration [40]. As discussed above, nNOS activates the canonical transnitrosylation cascade and simultaneously generates the non-canonical phosphorylation pathway through the activation of the sGC-dependent PKG axis. Hence, the inhibition of the nNOS activity could not distinguish which cascades regulate neurogenesis. More direct evidence focused on the S-nitrosylation in neuronal development has been reported. Myocyte enhancer factor (MEF) 2 contains a MADS box that binds to the CArG (CC[A/T]_6_GG) box to promote transcription and embryo development. The biological relevance of MEF2 is well-established not only in heart/muscle development, but also in brain development. In adult neurogenesis, the timed expression of nuclear receptor tailless (TLX) is an essential regulator. Several studies have found that MEF2, especially the MEF2A subtype, facilitates TLX transcription. As expected, MEF2 is recruited on the CArG box on the TLX promoter. Cys 39 at the MADS box underwent nNOS-dependent S-nitrosylation, which disrupted the pocket structure of the MADS box and subsequently interfered with DNA binding [41]. A somewhat opposite function of nNOS-dependent S-nitrosylation in neurogenesis has been reported. Tropomyosin receptor kinase B stimulates PSD95-coupled nNOS in response to the brain-derived neurotrophic factor. Cytoplasmic nNOS S-nitrosylates glyceraldehyde 3-phosphate dehydrogenase (GAPDH) to SNO-GAPDH, and promotes the nuclear targeting of SNO-GAPDH. SNO-GAPDH is utilized as a robust NO source for nuclear proteins B23, DNA-activated protein kinase, sirtuin-1, and histone deacetylase (HDAC) 2. BDNF-derived nitrothiol finally cascades S-nitrosylates HDAC2, C262, and C274 [42]. The S-nitrosylation of HDAC2 results in dissociation from the chromatin, which causes histone H3 and H4 acetylation. HDAC2 is involved in general transcription repression as an essential member of the nuclear repressor complexes, such as that of Sin3, NuRD, and CoREST [43]. Hence, the S-nitrosylation of HDAC2 finally potentiates transcription. Indeed, HDAC2 S-nitrosylation is mandatory for dendrite growth, branching, and neuronal migration. HDAC2 S-nitrosylation-dead mutant transduced cells accumulate in the intermediate zone in the mouse embryonic brain. HDAC2 S-nitrosylation is crucial for radial neuron migration, thus, for normal cortex development [44]. Collectively, the overall role of nitrosation in neuronal development is not simple to summarize. For adequate development, the fine regulation of S-nitrosylation in a spatiotemporal manner is necessary.

#### 2.1.2. Neurodegenerative Diseases and S-Nitrosylation

Although nitrosation modulates neurogenesis in a diverse way, nitrosative stresses consistently deteriorate neurodegenerative diseases. Besides the basal function of nNOS in neurons, iNOS notably contributes towards NO generation in the neuroinflammation process [45]. The source of iNOS is somewhat questionable. iNOS predominantly exists in immune cells, and regulates differentiation and the higher function of those cells. The major source of iNOS in the brain is residential microglia, but infiltrated monocytes and activated astrocytes are also considerable suppliers [46]. Many studies have pointed out that the aberrant activation of iNOS exerts the chronic neuroinflammation process; however, whether iNOS solely generates the neurotoxic amount of NO or whether nNOS-mediated nitrosative stresses contribute to neuronal degeneration remain to be elucidated.

It is common that ROS and reactive nitrogen species (RNS) are generated simultaneously and participate in neuronal cell damage and death, which results in neurodegenerative diseases [47]. Hence, it is difficult to consider the isolated role of nitrosation stress in neuroinflammation; however, cumulative studies have clarified the molecular events that are regulated by protein nitrosation. In a physiologic environment, protein nitrosylation finely regulates either synaptic function, through processes such as those involving the NMDA receptor and cation channels, or synaptic plasticity. The neurotoxic amount of NO substantially nitrosylates various substrates, which exacerbates mitochondrial dysfunction, protein misfolding, and, in turn, neuronal death.

As a more general modulator, the S-nitrosylation of GADPH has been extensively elucidated [48]. Atypical NOS activity, thus, substantial NO production, mainly takes place in the cytoplasm. Intriguingly, extensive S-nitrosylation also occurs, in which proteins are restricted in the nucleus. This observation raises two possible mechanisms regarding the source of NO for nuclear protein: the simple diffusion of NO into the nucleus or transnitrosylation by shuttling the protein. Based on multiple studies, transnitrosylation by S-nitrosylation protein redistributed from the cytoplasm to the nucleus is more widely accepted. Among the transnitrosylase of the nuclear substrates studied, GAPDH is the main contributor. Cytoplasmic GAPDH is primarily S-nitrosylated by nNOS, iNOS, or a robust NO donor at Cys 150, which augments the physical interaction with siah E3 ubiquitin protein ligase 1 (SIAH1). SIAH1 preferentially binds to S-nitrosylated GAPDH and elicits nuclear shuttling as a functional complex with SNO-GAPDH. Finally, nuclear SNO-GAPDH supplies NO to its substrate and, in turn, initiates the apoptosis of neuronal cells. Either reducing the S-nitrosylation of GAPDH or interfering with SIAH1 binding attenuates neuronal cell death even at the neurotoxic level of NO. GAPDH S-nitrosylation is also a notable modification to be addressed for the alleviation of neurodegeneration [49].

##### Alzheimer’s Disease

Neurons are highly dependent on mitochondrial respiratory chains because they expend huge amounts of energy. Hereditary disease that results in mitochondria dysfunction generally arises from neurodegenerative symptoms at younger ages and is increasingly aggravated in an age-dependent manner [50]. To maintain the healthy integrity of the total mitochondria, mitochondria dynamically undergo fusion/fission processes. In response to higher energy consumption, two or more individual mitochondria fuse together and thereby generate larger-sized mitochondria. On the other hand, accumulated damage stimulates segregation into two distinct mitochondria; one healthy and one damaged, which allows for the clearance of dysfunctional mitochondria [51]. This dynamic regulation to support energy demands and maintain healthy integrity is finely controlled by fusion (mitofusin and opa1) and fission proteins (fis1 and Drp1). Multiple studies have demonstrated that the functional relevance of nitrosation in Dynamin-related protein 1 (Drp1) contributes to the development of neurodegenerative disease, especially Alzheimer’s disease. Drp1 forms a homodimer through the N-terminal GTPase domain and again oligomerizes to generate a ring structure through fission. A considerable increase in NO levels in neuronal cells results in the fragmentation of the mitochondria, which suggests that NO accelerates the abnormal fission process. Either through treatment of the general NO donor Aβ25–35, or of the amyloid precursor protein, the S-nitrosylation of Drp1 is commonly observed. The S-nitrosylation of Drp1 is abolished when C644 is substituted with alanine. The S-nitrosylation of Drp1 promotes homodimerization but not polymerization, and acquires a higher GTPase activity. S-nitrosylation in Drp1 seems to enter the intermediate state to form a SS bridge for the homodimer, which exacerbates abnormal fission in neuronal cells [52].

##### Parkinson’s Disease

Accumulated misfolding proteins are highly toxic and fatal components of progressive neurodegeneration. Amyloid β, Tau, and α-synuclein are well-known proteinopathy-associated molecules in degenerative disease. In both patients with Parkinson’s disease with diffuse Lewy bodies and those with Alzheimer’s disease, the brain tissue shows the S-nitrosylation of protein-disulfide isomerase (PDI), which was phenocopied in rotenone-treated SH-SY5Y cells, a cellular model of Parkinson’s disease. S-nitrosylation of PDI interrupts normal protein folding and results in the accumulation of misfolding proteins. Through the cellular clearance system, polyubiquitination machinery are activated in response to aggregates and the sustained activation of the unfolded protein response pathway finally stimulates endoplasmic reticulum stresses. Consequently, chronic nitrosative stresses impair PDI function by S-nitrosylation, and the impaired clearance of misfolded proteins exacerbates neuronal cell injury and cell death [53].

#### 2.1.3. Epilepsy and S-Nitrosylation

Besides chronic neurodegenerative disorder, the pathophysiologic role of nitrosation stress in seizures has been reported. Epilepsy, a disease state of seizures, is due to repeated, uncontrolled, and propagation to neighbors of abnormal firing that results in focal or global excitation of the brain. In terms of aspects of individual subjects, patients with seizures that affect the motor cortex are susceptible to physical injury, such as abrupt falling, fractures, and even traffic accidents. At the microscopic level, sustained and prolonged excitation elicits the accumulation of neuronal damage and, finally, results in cell death [54]. The kainate receptor, specifically, the glutamate ionotropic receptor kainate type subunit 2 (GluK2), undergoes S-nitrosylation in an animal model of epilepsy. In response to agonist binding, the receptors allow for the influx of cations. Intracellular calcium performs diverse roles, such as activation of PLC, secondary calcium release, vesicle exocytosis, and nNOS activation. An aberrant increase of NO by nNOS primarily nitrosylates cytoplasmic proteins. GluK2 is also S-nitrosylated by kainite stimuli and S-nitrosylation of GluK2 further potentiates calcium influx. The S-nitrosylation of GluK2 accelerates the generation of the SNO-GluK2/PSD95/nNOS complex, which causes neuronal damage. Pharmacological inhibition of S-nitrosylation ameliorates epilepsy-induced cell death [55]. During epilepsy-mediated cell death, membrane targeting of the P2X7receptor plays a crucial role. Under pilocarpine-induced status epilepticus, PDI is initially S-nitrosylated and SNO-PDI transnitrosylated to the intracellular P2X7 receptor. Two S-nitrothiols in the P2X7 receptor form a secondary SS bridge, which maturates the P2X7 receptor structure and localizes in the membrane. Neighboring microglia and astrocytes recognize the P2X7 receptor with the SS bridge, and neuroinflammation processes are generated [55].

#### 2.1.4. Autism Spectrum Disorder and S-Nitrosylation

Autism spectrum disorder (ASD) is a neurodevelopmental syndrome that refers to the impairment of social interactions, communications, and repeated activities. The SHANK3 mutation in humans develops a different level of neuropsychiatric disabilities, including ASD, intellectual disorder, and Phelan–McDermid syndrome. Shank3 ablation phenocopies human ASD in mice; hence, Shank3 knockout (KO) mice are utilized for ASD studies. Amal et al. generated InsG3680(+/+) mice to mimic human mutations in mice and measured S-nitrosylation profiles. In mutant mice, S-nitrosylation of calcineurin elicits the phosphorylation of CREB and synapsin1, which, in turn, facilitate transcription and synaptic vesicle release, respectively. Increased GSNO rather than nNOS contributes to massive nitrosative stress [56]. Despite the diversity of animal models available for related studies, the overall findings of nitrosation in neurodevelopmental or neurodegenerative disease are somewhat redundant. RNS are generated in response to inflammatory stimuli, such as genetic abnormalities, chemical exposure, and chronic accumulation of misfolded protein and the nitrosase substrate directly or indirectly by transnitrosylation, which deteriorates disease progression. Developmentally appropriate S-nitrosylation finely regulates neurogenesis in a spatiotemporal manner, but S-nitrosylation mainly contributes towards neurodegenerative disease. Denitrosylation or the inhibition of nitrosylation as a strategy would be a considerable therapeutic target, at least to mitigate neurodegeneration in adult subjects.

### 2.2. The Immune System and Its Associated Disease

Even though several studies have elucidated that nNOS and eNOS contribute towards immune cell function [57,58], it is widely accepted that iNOS mainly regulates those processes. iNOS somewhat differs from the other subtypes (nNOS and eNOS). iNOS is expressed in response to exogenous stimuli or cytokines, whereas nNOS and eNOS are constitutively expressed. nNOS and eNOS produce NO in the presence of an intracellular calcium concentration; however, iNOS does not require calcium for its activity. iNOS is the predominant subtype of most immune cells and is essential for the development, maturation, and differentiation of T cells, B cells, monocytes, and dendritic cells.

#### 2.2.1. T Cells

A previous study using iNOS KO mice found that the loss of iNOS accelerates TH17 cell differentiation without notable changes in the Th1 or Th2 cell numbers or own functions. The treatment of N6-(1-iminoethyl)-1-lysine dihydrochloride, an iNOS-selective inhibitor, to CD4+ T cells significantly potentiates interleukin (IL)-17 production. In contrast, S-nitroso-N-acetylpenicillamine, a nonspecific NO donor, suppresses IL-17 generation in cultured T cells. The authors identified tyrosine nitration on RORγt for iNOS-dependent IL-17 expression. Tyrosine-nitration of RORγt interferes with binding on the promoter and thereby impairs the activation of IL-17 transcription, which plays a crucial role in immune modulation in an experimental colitis model [59]. Besides IL-17, IL-12 is also regulated by iNOS. The total IL-12 level is significantly increased in iNOS null mice, while robust NO donor S-nitroso-N-acetylpenicillamine strongly suppresses lipopolysaccharide (LPS)-mediated IL-12 production. NO attenuates LPS-induced IκB phosphorylation and NF-κB binding on the IL-12 promoter, which disturbs the transcription activity of NF-κB [60]. Although the authors did not clarify the relevant target molecule or target residue on NF-κB, nitration modification, such as tyrosine nitration or S-nitrosylation of both IκB and NF-κB, is involved in these pathophysiologic mechanisms. Besides helper T cell differentiation, it seems that nitrosative stresses also regulate T cell maturation and expansion. During final maturation, naïve T cells undergo selection in the thymus. Epithelial and dendritic cells in the thymus highly express iNOS in a constitutive manner and recognize CD4+CD8+ double-positive thymocyte cells. This cell-to-cell interaction with double positive cells and thymus cells via auto- or allo-antigens allows for apoptotic cell death, whereas single positive cells, CD4+CD8− or CD4−CD8+, escape from this negative selection [61]. A high level of iNOS produces a cytotoxic amount of NO, and those substantial nitrosative stresses generate ONOO^−^, which induces apoptosis. On the other hand, the expression of constitutive NOS (nNOS or eNOS) results in a salvage pathway in the negative selection of double positive cells. The suppression of CD95L expression and/or caspase nitrosation by nNOS or eNOS may contribute to the proliferation of single positive T, CD4+, and CD8+ cells.

#### 2.2.2. B Cells

The functional relevance of NOS in B cells is relatively limited compared with the other immune cells. There are two different distinct subtypes in plasma cells: terminally matured B cells and short- and long-lived plasma cells. Short-lived plasma cells have an average life span of less than one month and mostly contribute to the acute defense system by producing antibodies. However, long-lived plasma cells persist throughout the host’s lifespan after settling in the bone marrow, and participate in host immunity. Therefore, maturation of long-lived plasma cells, and thus, their increased longevity during the host’s survival, are necessary for proper protection from exogenous infections. iNOS has been highlighted as a novel regulator of plasma cell survival. The lifespan of antibody-secreting cells from iNOS-null mice is relatively shorter than that for cells from wild-type (WT) mice. In vitro treatment with non-selective NO inhibitors in antibody-secreting cells from WT mice results in the acceleration of cell death, whereas the NO donor normalizes a shorter lifespan for those cells from iNOS-null mice. The survival benefits of antigen-secreting cells by IL-6 are absent in cells from iNOS2 KO mice, which implies that the iNOS-dependent nitrosation of the IL-6-response signal cascade plays a crucial role in plasma cell longevity and subsequent lifespan immunity [62].

#### 2.2.3. Macrophages

Most of all, the functional importance of NO and RNS is well-established in macrophage-mediated innate mechanism. Macrophages crudely refers to cells that are involved in the immune system through the engulfment and digestion of anything, including microbes, cellular debridement, foreign bodies, and even cancer cells [63]. The term macrophage mainly refers to a type of white blood cell that also exists in a variety of tissues, with different names such as Kupffer cells (liver), mesangial cells (kidneys), alveolar macrophages, microglia (brain), and osteoclasts (bones). Although a more detailed specification is more common at present, it remains effective to divide subtypes of macrophages into two distinct classifications: M1 and M2. Crudely, M1 macrophages contribute to the acute inflammation process, whereas M2 macrophages mainly resolve inflammation and participate in the tissue repair mechanism. The most reliable markers for distinguishing M1 and M2 macrophages are arginase and iNOS. Both arginase and iNOS catalyze arginine in different ways; arginase catalyzes arginine into L-ornithine and urea, whereas iNOS catalyzes arginine into citrulline and NO. Hence, the abundance of arginase in the cell results in the exhaustion of arginine, a substrate of NO. M1 macrophages, which show a high iNOS expression, produce huge amounts of NO and utilize NO/RNS to kill the microorganisms that they engulf [64]. Although the NO-dependent host defense mechanism is well-established, the iNOS-dependent nitrosation mechanism associated with macrophage function remains unclear in terms of the digestion of the microorganisms. Therefore, more detailed studies regarding this aspect are necessary.

#### 2.2.4. Autoimmune Diseases

Innate immunity is mostly associated with macrophages and their inflammatory response, while lymphocytes are involved in adaptive immune responses. Most inflammatory responses are an orchestration of innate and adaptive immune reactions. Hence, nitrosation regulates not only appropriate immune responses to protect the host, but also atypical autoimmune diseases in diverse ways.

##### Multiple Sclerosis

Encephalomyelitis disseminata, more commonly known as multiple sclerosis (MS), is an autoimmune disease that targets the blood−brain barrier and myelin sheath in the central nervous system [65]. Efferent and afferent signal conductions are retarded due to demyelination, affecting autoimmune responses. Depending on the dominantly affected area, patients complain of different symptoms, including diplopia (double vision), blurred vision, blindness, loss of general sensation, and ataxia with muscle weakness. Most MS cases tend to be abruptly onset by unknown triggers, and subjects suffer from recurrent MS attacks. Like different types of autoimmune disease, the underlying mechanism remains unknown. However, many studies have revealed reliable disease markers, such fetuin-A, osteopontin, IL, NO, and iNOS. Among the markers, the concentration of NO and its metabolite in the cerebrospinal fluid reflects disease severity [66]. The role of NO in MS is relatively well-established. Reactive microglia, astrocytes, and macrophages from the blood stream express iNOS and disgorge a deleterious amount of NO. NO/RNA and the subsequent nitrosation, in turn, ablate the blood−brain barrier, myelin, and axons through either a direct or indirect pathway. Demyelination or neuronal cell death shares common mechanisms for neurodegenerative disorders.

##### Rheumatoid Arthritis

Rheumatoid arthritis (RA) refers to destructive chronic autoimmune disease by which immune cells are activated against synovial fibroblasts and chondrocytes [67]. Several studies have revealed that nitrosation may contribute to RA development or progression. The total NO amounts are increased in the fluid from RA patient synovium or collagen-induced arthritis models. Gupta et al. compared the plasma samples from healthy subject controls and RA patients. They observed that the S-nitrosylation of mannose binding lectin, a robust complement activating molecule in the innate immune reaction, significantly increased in RA patients [68]. Caspase nitrosylation was reported as a detailed pathophysiology. In the initiation and progression of RA, RA fibroblast-like synoviocytes (FLS) deteriorated chronic inflammation and joint destruction. RA-FLS atypically survives even under the apoptotic cell death signal. In association with nitrosative stresses in the blood and synovial fluid, intraarticular environment molecules might also undergo nitrosation. Among the apoptosis signals, extensive studies have shown that the activated caspase signal is impaired. The death receptor Fas receptor conjugates pro-caspase as an intracellular anchor. When Fas ligands activate the Fas receptor, sequential caspase cleavage takes place and, in turn, cleaved and maturated caspase 3 executes the cell death program. S-nitrosylation on caspase 3 (Cys 163) [69] or caspase 9 [70], however, allows for resistance against the proteolytical cleavage process, which finally blocks the propagation of the death signal. General nitrosative stresses result in cell death, especially in neuron cells; however, S-nitrosylation in RA progression protects against cell death. This discrepancy in the final outcome after nitrosation needs to be clarified.

#### 2.2.5. Septic Shock

Beyond chronic autoimmune diseases, nitrosative stresses strongly accelerate the acute immune storm. Sepsis is an acute series of host responses to infection by microorganisms and to life-threatening conditions. Most of the initial process is localized to the infection site. Focal infection is sometimes propagated in the blood stream when the host suffers from an immunocompromised condition, such as that associated with cancer therapy, diabetes, and old age. Microorganisms disseminate throughout the body through the blood flow and stimulate massive immune reactions, which results in a fatal crisis. The roles of NO and NOS in sepsis have been reported in several studies thus far. An experimental sepsis model-based study revealed that LPS triggers iNOS expression and eNOS activation through pro-inflammatory cytokines, such as TNFα, IL-1, and IL-6. It seems that a profound surge of NO by iNOS/eNOS in the blood directly participates in hypotensive crisis accompanied with sepsis. High levels of NO and NRS widely disperse over multiple organs and nitrosate substrates [71,72]. Similarly, the accumulation of nitrosation stress in the cells provokes tissue damage.

#### 2.2.6. Miscellaneous

The pathophysiology and disease progression of many different kinds of life-threatening autoimmune diseases, such as systemic lupus erythematous and ankylosing spondylitis, remain unclear due to a lack of definitive animal models [73,74]. Although their affected organs and symptoms are quite different, the chronic inflammation whereby abnormally activated immune cells turn against the host is a common mechanism underlying these conditions; this implies that ROS/RNS may be involved in disease aggravation and in the permanent deformity of the host. The functional relevance of RNS and its implications must be elucidated.

### 2.3. The Cardiovascular System

Because of the potent vasodilation effect of NO, it is crudely regarded that NO has a beneficial role in the heart [75]. NO not only improves the peripheral blood flow and increases the O_2_ and nutrition supply through the regulation of arterial relaxation, but also decreases cardiac load as the heart requires less pressure to pump out blood. Like other diseases, RNS and ROS levels are also commonly increased in cardiac diseases. Aberrant oxidative species can induce nitrosative modification by an increase in NO, NO_2_, ONOO^−^, and GSNO. Sustained RNS for a longer period results in S-nitrosylation and tyrosine nitration. Interestingly, both S-nitrosylation and tyrosine nitration are also induced without notable alterations to the cellular NO level. To date, numerous evidence has supported its functional importance in cardiac hypertrophy, heart failure, ischemic heart disease, arrhythmia, and other heart diseases.

ROS and RNS cause contractile dysfunction and are more prominent in end-stage heart failure patients compared with non-failing hearts. In particular, S-nitrosylation in myofibril components is obvious in patients with heart failure. Among the numerous myofibril factors, tropomyosin is a well-known target for S-nitrosylation, carbonylation, and the SS cross bridge [76]. S-nitrosylation is also observed in dilated cardiomyopathy [77,78] and atrial fibrillation (AF) [79]. The S-nitrosylation of protein generates the deterioration of cardiac disease through the deterioration of abnormal protein function. Hence, targeted approaches to reverse protein S-nitrosylation can be a promising therapeutic target.

#### 2.3.1. Cardiac Ion Channels and Ion Homeostasis Proteins

##### L-Type Calcium Channel

Many cardiac diseases are closely associated with cardiac muscle contractility; cardiac muscle contraction is closely regulated by cytosolic calcium ions. Cardiac muscle contraction begins from β-adrenergic stimulation and thereby the activation of cyclic adenosine monophosphate/protein kinase A, which, in turn, leads to the accumulation of calcium ions in the cytosol via the opening of calcium channels and pumps. During muscle contraction, L-type calcium channels (LTCCs) located in the plasma membrane allow Ca^2+^ to enter the cytosol in response to β-adrenergic stimulation. Intracellular calcium binds to the sarcoplasmic reticulum (SR) ryanodine receptor (RyR), which results in a further release of Ca^2+^ from the SR into the cytosol. Finally, Ca^2+^ binds to the troponin complex, leading to myocyte contraction. Nitrosative modification in calcium ion channels and pumps occurs in pathological conditions with impaired muscle relaxation or contraction. Exaggerated stimulation of β-adrenergic receptors induces RyR2 S-nitrosylation through the nNOS activity. RyR2 S-nitrosylation increases its activity and even releases Ca^2+^ during diastole. High amounts of Ca^2+^ cause abnormal contraction during diastole, which leads to diastolic dysfunction [80,81,82,83,84,85]. Interestingly, it is suggested that RyR1, a skeletal muscle type of RyR, is excessively S-nitrosylated, oxidized, and phosphorylated. This result emphasizes that pathologic Ca^2+^ release is a potential mechanism behind skeletal muscle weakness and impaired exercise tolerance in patients with heart failure [86]. Taken together, impaired S-nitrosylation regulation of either RyR1 or RyR2 induces an abnormal cytosolic calcium level and, in turn, leads to contraction failure and falls in the pathological phase.

Initial Ca^2+^ influx through LTCC sequentially activates the RyR Ca^2+^ channel located in the SR. It has been reported that female hearts exhibit a resilience capacity to the injury with a significant increase in functional recovery compared with male hearts under ischemia/reperfusion (I/R) injury, and the S-nitrosylation of LTCC is closely linked to those beneficial effects. Females tend to express more abundant levels of eNOS when compared with males in I/R injury, which, in turn, induces the S-nitrosylation of LTCC, specifically α1 subunit (LTCCα1). S-nitrosylated LTCCα1 attenuates Ca^2+^ influx and elicits a protective role after I/R injury through the inhibition of excess stimulation [87].

Additional evidence supports that the S-nitrosylation of LTCC retards the calcium current (I_Ca,L_) in human disease. The L-type calcium current (I_Ca,L_) was decreased in the atrial specimens from AF patients. A biotin switching assay, which was performed with an AF patient sample, revealed that global S-nitrosylation was increased and the α1C subunit of the LTCC is one of the S-nitrosylation targets in association with AF. Besides the S-nitrosylation of LTCCα1, the total GSH content was significantly lower in AF patients compared with that in healthy human subjects with no AF history. As discussed, GSH is involved in the antioxidant process, which protects against oxidative and nitrosative stress and denitrosylates S-nitrosylated proteins. The incubation of primary cultured myocytes from AF patients with N-acetylcysteine, a precursor of GSH, successfully restores both the S-nitrosylation of LTCCα1 and, thereby, the L-type calcium current (I_Ca,L_) [79]. However, it remains controversial whether the S-nitrosylation of LTCC attenuates the calcium current (I_Ca,L_). One study suggested that SNAP, a NO donor, increases the Ca^2+^ current (I_Ca,L_). Right atrial appendages obtained from AF patients and a control subject who had a normal sinus rhythm showed no significant difference in S-nitrosylation. For the possible mechanism of these somewhat controversial findings, the dual effect of NO on cGMP-cAMP cross-regulation was identified. In the basal condition, NO stimulates sGC to produce cGMP and, in turn, increases the Ca^2+^ current (I_Ca,L_). However, NO also inhibits the Ca^2+^ current (I_Ca,L_) via cAMP stimulation, such as β-adrenoceptor activation, which suggests that channel S-nitrosylation in LTCC may contribute minimally to the regulation of the Ca^2+^ current (I_Ca,L_), or that S-nitrosylation at different cysteines may result in opposite functions, like the phosphorylation of GSK3. A more detailed study to delineate the responsible target cysteine and its specific role is necessary [88].

##### Sarco/Endoplasmic Reticulum Ca^2+^-ATPase

To perform the regular function, aberrantly increased levels of calcium, which is released during contraction, should be eliminated during diastole. The sarco-/endo-plasmic reticulum Ca^2+^-ATPase (SERCA) is an active transporter that transfers Ca^2+^ from cytosol into the SR and then withdraws Ca^2+^. The S-nitrosylation of SERCA2a and its implication was discovered through the use of the chemo-denervation rat model. In association with the reduced eNOS activity, ONOO^−^ generation and, thereby, tyrosine nitration were also decreased. Furthermore, SERCA2a S-nitrosylation was attenuated. The chemo-denervated rat showed increased ventricular end-diastolic pressure, a surrogate sign of diastolic dysfunction. In short, S-nitrosylated SERCA2a performs a physiological function. Specifically, myocardial relaxation and the reduction of SERCA2a S-nitrosylation cause diastolic dysfunction [89]. One study further supported this hypothesis by finding that the SERCA S-nitrosylation level was positively correlated with the recovery rate in an I/R injury model [90].

Taken together, S-nitrosylation on calcium channels finely regulates the channel activity and performs a physiological function. The S-nitrosylation of either SERCA or LTCC lowers the cytosolic Ca^2+^ increase, while the S-nitrosylation of RyR accumulates cytosolic Ca^2+^. Thus, it is important for developing modality to alleviate ischemic disease by inducing the S-nitrosylation of LTCC or SERCA, or the diastolic dysfunction by the de-nitrosylation of RyR. S-nitrosylation regulation on the Ca^2+^ channels is suggested as a potential target for pharmacologic intervention.

#### 2.3.2. Arrhythmia

Various cation channels, including LTCC, undergo S-nitrosylation modification in cardiac arrhythmia. An ex vivo arrhythmia model using carbon monoxide revealed that arrhythmia arises from NOS activity. NO causes the S-nitrosylation of Na(v)1.5, which induces late Na+ inward currents (*I_Na_*) and thereby early after-depolarization-like arrhythmias [91]. Supportively, the administration of NOSi successfully reverses the aberrant Na+ inward current: L-NAME [91] or L-NMMA [92]. Na(v)1.5 forms a macromolecule complex with nNOS and caveolin-3. Late *I_Na_* is significantly altered depending on the caveolin-3 genotype. The mutant variant caveolin-3 fails to inhibit nNOS activity and facilitates the S-nitrosylation of Na(v)1.5, which elicits sustained late sodium current *I_Na_* and long QT syndrome [92]. The potassium channel is also a target of S-nitrosylation. The potassium channel Kir2.1 that participates in the inwardly rectifying current, *I_K1_*, is finely regulated by S-nitrosylation in physiologic conditions. The NO donor induces both S-nitrosylation at Kir2.1 Cys 76 and atrial *I_K1_*. Human samples from chronic AF showed lower levels of Kir2.1 S-nitrosylation. Impaired NO generation might fail to make *I_K1_* and thus develop arrhythmic changes [93]. Decisively, de-nitrosylation or desensitization of the cation channel can be a novel target for treatment associated with cardiac disorder, especially atrial arrhythmias.

The role of protein nitration in chronic arrhythmia-induced heart failure has also been reported. Direct pacing in the ventricle for 3 weeks successfully induced systolic heart failure and iNOS is dramatically increased in a failing heart. iNOS generates both NO and ONOO^−^, which implicates the uncoupling of iNOS, and more likely induces protein nitration rather that S-nitrosylation. The inhibition of iNOS with 1400 W attenuated superoxide anion formation and improved contractile dysfunction in failing hearts [94]. On the other hand, it was also reported that nitrosative stresses minimally affect heart failure development. Although continuous right ventricular pacing successfully generated systolic heart failure and global left ventricular remodeling in a female large white breed swine model, tachycardia-induced HF swine with impaired ventricular function had more considerably decreased iNOS mRNA and protein levels than those of the sham swine. S-nitrosylation was decreased in parallel with pacing duration and heart failure severity [95]. This finding is somewhat controversial compared with other results that state that the NOS activity is commonly correlated with disease severity. However, estradiol is a known activator of the TRX system. Hence, the discrepancy of the results is possibly from sex differences.

#### 2.3.3. Dilated Cardiomyopathy

Diverse studies have suggested that nitrosative stresses exacerbate diabetic cardiomyopathy. Diabetic cardiomyopathy models, which are induced by either a high-fat diet or streptozotocin injection, specifically induce iNOS expression. Increased iNOS expression induces both oxidative and nitrosative stress, which results in S-nitrosylation and tyrosine nitration. Interestingly, the serum nitrite and nitrate levels are not altered in diabetic mice despite robust RNS changes. Tetrahydrobiopterin (BH4) is an essential cofactor of NOS that governs NOS coupling and subsequent NO generation, especially for iNOS. Treatment with BH4 increased NO bioavailability and protein S-nitrosylation in diabetic rats, which emphasized that iNOS uncoupling is a crucial defect in diabetes-induced cardiomyopathy. More directly, resolving iNOS uncoupling with BH4 treatment contributes towards tolerance to I/R injury and reduces the scar size in the hearts of diabetic rats [96]. Diabetic cardiomyopathy commonly accompanies cardiomyocyte death, and nitrosative modification controls such types of apoptotic cell death. Chemical-induced diabetes also elicited iNOS expression, S-nitrosylation, tyrosine nitration, and irreversible hyperglycemia in a rat model. Nitrosative stress S-nitrosylates GAPDH and, in turn, facilitates the interaction with Siah1. The SNO-GAPDH-Siah1 complex shuttles into the nucleus to activate nuclear apoptotic factors, then promotes cardiac cell death [97]. Strikingly, exercise seems to neutralize the aberrant activity of iNOS. iNOS inhibitors, 1400 W, and exercise successfully reduced the infarct size in diabetic mice, and even in mice with a defective eNOS pathway. This study implies that nitrosative stress, especially that generated by uncoupling iNOS, mainly exacerbates cytotoxicity, and physical exercise could alleviate nitro-oxidative stress, such as that caused by iNOS inhibitors [98]. Taken together, the iNOS expression is increased in diabetic cardiomyopathy and leads to nitrosative stress, and thus protein nitrosation. Increased S-nitrosylation in diabetic hearts is strongly associated with cardiac cell death. However, the recovery of uncoupled iNOS can improve nitrosative stress and minimize ischemic damage.

#### 2.3.4. Ischemic Heart Disease

Ischemic heart events are a fatal crisis that frequently result in sudden cardiac death. Myocardial infarction is commonly followed by this event and it is still the most common course of systolic heart failure. Hence, it is crucial to minimize the infarction area in ischemic heart attack. Several studies support that nitrosation can contribute to the alleviation of myocardial infarction or I/R injury in association with ischemic preconditioning. In a preconditioned heart, S-nitrosylation protein levels predominantly increase in the mitochondria. AKT/eNOS signaling is activated and allows eNOS translocation to the mitochondria. eNOS in mitochondria S-nitrosylates respiratory chain complexes. Non-selective NOSi, L-NAME, abolishes the cardio-protective effect of ischemic preconditioning, which suggests the beneficial role of eNOS in cardiac physiology [99]. This positive evidence of S-nitrosylation and NO in myocardial protection was further supported by neoangiogenesis through hypoxia inducible factor-1α (HIF1α). GSNOR KO mice confirmed that nitrosation in the heart reduced infarct scarring and preserved both ventricular systolic and diastolic function. These mice had abundant vasculature and sufficient tissue oxygenation when compared with WT mice. In GSNOR KO mice, HIF1α S-nitrosylation was significantly increased and that modification enhanced its binding to the promoter region of the vascular endothelial growth factor (VEGF) gene and, finally, stimulated neoangiogenesis [100].

Cardioprotective protein was also S-nitrosylated and was retained in ischemic preconditioning hearts. The tripartite motif-containing protein 72 (TRIM72) functions as a cardio-protective membrane repair protein. Irreversible oxidation by hydrogen peroxide (H_2_O_2_) leads to TRIM72 degradation, which takes place in oxidative stress-induced cell death [101]. GSNO induces TRIM72 S-nitrosylation at Cys 144 and leads to resistance against oxidation-induced degradation in an ischemic preconditioning heart [102]. TRIM72 S-nitrosylation dead mutant with C144 mutated to a serine reduced infarct size in ischemic mouse hearts. Taken together, oxidative modification at Cys 144 results in the degradation of TRIM72 and S-nitrosylation blocks fatal oxidation [101]. This observation could be an explanation for one of the competitive modifications that occurs in a single target amino acid, such as acetylation, methylation, and ubiquitination in a single lysine side chain.

#### 2.3.5. Heart Failure with Preserved Ejection Fraction

Heart failure commonly refers to reduced contractile function. Most patients experience shortness of breath, exercise intolerance, and even resting dyspnea. An ischemic heart event and thereby a loss of effective myocardium is a well-known pathophysiology of heart failure. This kind of heart failure characterizes predominant impairment of the ejection function, which is commonly less than 40% in echocardiogram evaluation. In the last two decades, however, the number of patients that complained of failing heart symptoms with normal ejection fraction has steeply increased. Because this group of patients show a well-conserved ejection fraction, physicians diagnose these patients with heart failure with preserved ejection fraction (HFpEF), while the classical concept of heart failure is named as heart failure with reduced ejection fraction (HFrEF). Although HFrEF has a well-organized management strategy depending on the classification or severity of the failing heart symptoms, no therapeutic modality is available for HFpEF. Although large animal studies to delineate the pathophysiology and clinical trials with empirical approaches through the use of established regimens for HFrEF have been undertaken, neither the mechanism of pathogenesis nor clinical trials have been successful. Disappointingly, all trials with NO donors, such as nitroglycerin, mononitrate, and dinitrate, have failed. Recent studies, including ours, could help explain why NO donor studies have failed.

Our group introduced a hypertension-associated diastolic dysfunction mouse model and a human heart with presumed HFpEF. The mouse model showed significantly increased blood pressure with impaired diastolic function and exercise time. Both the mouse model and human heart revealed that global S-nitrosylation was dramatically increased, predominantly in the nucleus fractions. Among NOS, the nNOS level was specifically increased and S-nitrosylated several target genes, including GAPDH and HDAC2. The targeted point mutation of HDAC2 Cys 262 and Cys 274 completely abolished the S-nitrosylation in response to diastolic dysfunction stresses, and the diastolic function remained normal. Indirect denitrosylase nuclear factor erythroid 2-related factor 2 (NRF2) successfully removed HDAC2 S-nitrosylation and dimethyl fumarate (DMF), a putative NRF2 inducer, reverse diastolic dysfunction, and HDAC2 S-nitrosylation [78]. Our study explained that specific protein S-nitrosylation aggravates HFpEF, but still failed to clarify in detail the mechanism of how that modification accelerates disease progression. However, we demonstrated that the in vivo denitrosylation strategy is effective for HFpEF intervention.

More importantly, another HFpEF preclinical study was reported by utilizing a combination of comorbidities of HFpEF: obesity and hypertension. Mice were provided with a high-fat diet and L-NAME to induce obesity/metabolic syndrome and hypertension by eNOS inhibition, respectively. The combination diet specifically induced iNOS expression in the heart and S-nitrosylation of inositol-requiring protein 1α (IRE1α). Even though NOSi L-NAME was used, S-nitrosylation was increased by iNOS activation. S-nitrosylation of IRE1α attenuated phosphorylation and its activity, which culminated in defective XBP1 splicing [103]. This study and ours show that NO-dependent modification in the heart participates in disease progression, and the denitrosylation or inhibition of nitrosative stresses more decisively improves HFpEF.

#### 2.3.6. Aortic Dissection

An interesting study showing S-nitrosylation and human disease was recently reported. Pan et al. observed the S-nitrosylation of plastin-3 in patient specimens from thoracic aortic dissection. Angiotensin II treatment in human umbilical vascular endothelial cells produced plastin-3 S-nitrosylation and thereby endothelial dysfunction. S-nitrosylation at plastin-3 Cys 566 by iNOS deteriorated adherens junction disorganization, which impaired the formation of intact vasculature. A loose vessel structure possibly contributed toward aortic rupture or dissection [104].

### 2.4. Tumor Microenvironment

The accumulation of somatic mutations, which has benefits for survival, finally converts to problematic cells. These cells bin to grow abnormally, claim the blood supply, and incapacitate the immune system, which, in turn, evolves into malignant cells. Numerous studies have elucidated the functional importance of nitrosative stress in the pathogenesis of cancer cell development. For example, there is a distinct phenotype wherein the iNOS level is aberrantly increased in many solid tumors and is positively correlated with the nitrosation of p21(ras). The S-nitrosylation of p21(ras) elicits an oncogenic progression through Ets-1 transcription in triple negative breast cancer [105]. Malignant cells often require high energy and O_2_ supplies. Relatively high demands result in hypoxic insult, which transactivates the eNOS/iNOS expression and stabilizes HIF1α. NO transiently relaxes blood beds to secure hypoxia, while HIF1α transcripts a proangiogenic factor, such as VEGF, for neoangiogenesis. The S-nitrosylation of HIF1α at two cysteines grants the dominant function; S-nitrosylation at Cys 520 (Cys 533 in mouse isoform) interferes with the binding of the E3 ligase and protects against degradation. S-nitrosylation at C800 (Cys 810 in the mouse isoform) facilitates the generation of functional complexes, such as p300 and CBP, which activate transcription [106].

Normally, the host’s immune system can recognize malignant-transformed cells and remove them from the body. Strikingly, cancer cells aggressively disturb the immune system rather than govern the cells to get support [107]. Although the immune system functions normally, cancer cells escape the immune response. The term tumor microenvironment indicates this confused situation. Many heterogeneous groups of cells participate in the tumor microenvironment: cancer cells and the surrounding cells, including immune cells, vascular beds, fibroblasts, and the extracellular matrix. Nitrosation is widely involved in the tumor microenvironment.

In response to tumor progression, many immune cells gather in cancer environments [107]. The location of the original source remains unclear, and most of the population of macrophages migrate from the bloodstream and markedly accumulate around the cancer cells. Initially, macrophages are attracted to cancer cells, but they form tumor-associated macrophages (TAMs). TAMs are a relatively undifferentiated population that lack a typical phenotype that can be used to distinguish immunostimulatory (M1) or immunosuppressive (M2) macrophages; however, it is generally accepted that TAMs share the M2-like phenotype, even when the TAMs remain in the undifferentiated state. TAMs are totally subordinate to cancer cells, and support them in escaping cytotoxic clearance and proceeding with distance metastasis. Recent studies have revealed the relationship between protein nitrosation and TAMs. Aberrant upregulation of iNOS/eNOS from cancer cells produces a large amount of NO, which induces the nitrosation of TAMs. Among the numerous target substrates of nitrosation, the S-nitrosylation of NF-κB is crucial to acquire the paradoxical function of TAMs. S-nitrosylation takes place in numerous NF-κB signaling molecules, and both p50 and p65 subunits, which directly promote transcription, are critical targets. S-nitrosylation in p50 (Cys 62) and p65 (Cys 38) attenuates the DNA binding affinity, which, in turn, fails to activate the transcription of proinflammatory cytokines, including interferon-γ and IL-1. High levels of NO can modulate cytotoxic T cell function in the tumor microenvironment. A high concentration of NO allows for the S-nitrosylation of the signal cascade to generate effector and memory T cells. NO directly inhibits the activity of Janus kinase 3, early response kinase, and protein kinase B, which prevent the IL-2 response [108]. Furthermore, cytotoxic T cell recruiting chemokine CCL2 is the target of nitration in the tumor microenvironment. Unfortunately, nitration/nitrosylation interferes with the infiltration of cytotoxic T cells. The chemical ablation of nitration enhances intratumoral T cell infiltration and the survival of the experimental mice [109]. As discussed above, NO and RNS are highly toxic molecules that are used for the innate immune defense. Continuous exposure allows NO and RNS to stimulate the S-nitrosylation of arginase 1. Arginase 1 S-nitrosylation leads to the consumption of L-arginine, the main source of NO, which finally impedes cytotoxic clearance by macrophages and accelerates the M2 polarization of TAMs. Collectively, nitrosative stresses accumulate in cancer cell environs and dysregulate the natural function of each component cell. Ironically, the nitration of target proteins renders survival benefits to cancer cells and supports cancer cell malignancy.

### 2.5. Miscellaneous Diseases

#### 2.5.1. Diabetes Mellitus

As discussed, NO and RNS are generally accompanied with ROS levels that are commonly increased in chronic inflammation. Hence, nitrosative stresses largely accumulate in other organs that have undergone chronic inflammation or metabolic disorder. Diabetes mellitus refers to high blood sugar by through of insulin or resistance against insulin-sensitive organs. High amounts of sugar in the blood tend to induce glucotoxicity and lipotoxicity, which produce ROS and RNS. eNOS has to form a functional complex that includes a homodimer to produce NO. However, red blood cells [110] and the endothelium [111] exposed to hyperglycemia preferentially generate ONOO^−^ instead of NO because of eNOS uncoupling. S-nitrosylation at eNOS Cys 443 impedes the dimerization and coerces it to remain a monomer. The eNOS monomer produces ONOO^−^ instead, which amasses nitrosative stress and aggravates endothelial dysfunction [112].

Diabetic hyperglycemia also directly targets the kidneys. The kidneys filter blood to clear metabolites and toxic materials from the bloodstream. For this reason, kidneys are commonly affected by chronic inflammation. The natural source of nitrosative stress is unclear, but robust S-nitrosylation is observed in an experimental model of diabetic nephritis. S-nitrosylation is positively correlated with disease severity [113]. As a limited number of studies have been reported, the functional relevance of nitrosative stress in chronic renal disease, such as acute kidney injury, chronic renal failure, and nephrotic syndrome, has to be elucidated.

#### 2.5.2. Chronic Hepatitis and Hepatocellular Carcinoma

The liver is also a common organ that can develop chronic inflammation through viral infection, alcoholic liver disease, non-alcoholic hepatosteatosis, etc. No obvious subtype of NOS exists in hepatocytes. Hence, the most responsible subtype for nitrosative stress is iNOS, whereas eNOS primarily protects hepatocytes [114]. Chronic viral hepatitis by hepatitis B and C viruses is well-established as the common cause of hepatitis, fibrosis, and hepatocellular carcinoma. Interestingly, mouse experiments with iNOS KO revealed that the aberrant increased iNOS expression in hepatitis B virus infection switches the metabolic pathway through the upregulation of enzymes involved in gluconeogenesis [115]. Unfortunately, chronic infection by hepatitis B virus results in the development of hepatocellular carcinoma. Cumulative evidence suggests that iNOS-mediated nitrosative stress is frequently augmented by GSNOR deficiency, which mainly modulates intracellular S-nitrosylation through the reduction of GSNO [116,117]. About 50% of hepatocellular carcinoma patients lose GSNORs. Microdeletion flanking the GSNOR legion is frequently observed in hepatocellular carcinoma patients [118].

In a tight association with iNOS, the loss of GSNORs notably synergistically increases S-nitrosylation and accelerates the degradation of the DNA repair protein O6-alkylguanine-DNA alkyltransferase, which profoundly contributes towards hepatocellular carcinoma development. Recently, the role of S-nitrosylation in the progression of non-alcoholic steatohepatitis has been reported [119]. A methionine choline-deficient formula was provided to induce non-alcoholic steatohepatitis, and the S-nitrosylation of peroxisome proliferator-activated receptor γ was observed, which deteriorates Kupffer cells and, in turn, participates in non-alcoholic steatohepatitis. It is noteworthy to consider the role of leptin in hepatosteatosis. The level of leptin increases in parallel with insulin resistance and modulates iNOS expression and nicotinamide (NAM) adenine dinucleotide phosphate (NADPH) oxidation, which accelerates both Kupffer cell activation and tyrosine nitration [120,121]. The role of nitrosative stresses in hepatitis largely remains as circumstantial evidence. Therefore, more detailed study is necessary. The functional relevance of nitrosation in human disease is summarized in Figure 1.

## 3. Reducing Protein Nitrosation and Its Therapeutic Potential

Protein S-nitrosylation has a reversal modification called denitrosylation. Like the S-nitrosylation mechanism, denitrosylation is also accomplished through the transnitrosylation process. Reduced GSH and thioredoxin (TRX) function as denitrosylases [122].

### 3.1. Glutathione and S-Nitrosoglutathione System

GSH recognizes and binds to the S-nitrosylated protein, which chelates NO from the original protein and finally forms GSNO. Of course, GSH also captures free NO in the cell and stores NO as GSNO. As discussed already, GSNO serves as a physiologic NO donor in the cell. Exhausted GSH can be restored by the reducing process of GSNO. GSNOR, also known as aldehyde dehydrogenase 5, catalyzes GSNO and generates the temporal, unstable molecule N-hydroxysulfinamide (GSNHOH) in the presence of NAM adenine phosphate (NADH). GSNHOH intermediate metabolites, in turn, conjugate with GSH and convert to GSSG. GSH reductase (GR) cleaves GSSG into two GSHs with NADPH. Because of its unstable features, GSNHOH is spontaneously transformed to GSH sulfonamide and GSH SO_2_H in the absence or at a lower level of GSH. Chronic or massive oxidative stress can exhaust GSH, which further accelerates the depletion of GSH and thereby the aggravation of oxidative stress. It is not definite, but GSNOR appears to preferentially utilize NADH for reducing GSNO, whereas GR utilizes NADPH more frequently [123].

### 3.2. Thioredoxin and the Thioredoxin Reductase System

The other major system to reduce S-nitrosylated protein is TRX. There are two main subtypes of TRX in mammalian cells: TRX1 mainly localizes in the cytoplasm and TRX2 distributes in the mitochondria. In addition, the truncated isoform TRX80 has been reported, but its function is quite different from the original mature forms [124]. TRX1 and TRX2 share the active motif Trp-Cys-Gly-Pro-Cys, and their major redox property is mediated by two cysteines. Like transnitrosylation by GSH to GSNO, TRX induces transnitrosylation and turns into SNO-TRX, while the S-nitrosylated protein is reduced. At different points in the GSNOR system, S-nitrosylated TRX finally forms an intramolecular SS bond and simultaneously releases free NO or nitroxyl (HNO), depending on the S-nitrosylation amount in the TRX molecule. TRX reductase (TrxR) breaks the SS bonds in TRX in the presence of NADPH. Interestingly, TRX does not catalyze NO, whereas GSNOR metabolizes NO into hydroxylamine. More importantly, TRX can specifically break down the intramolecular or intermolecular SS bridge. As discussed, S-nitrosylation partially contributes towards SS formation. SS bonds are critical for proper folding, and thus for the maturation of protein; however, an abnormal intramolecular SS bond alters the 3D structure and its own function. Furthermore, a nonspecific intramolecular SS bridge can form an abnormal complex or even generate aggregates. Hence, the proper activity of TRX in the cell is necessary to maintain mature function. Freely released NO can be reutilized for intracellular NO physiology and signaling. Intriguingly, HNO is also released by TRX. Although the functional importance of HNO remains unclear, several studies have clarified that HNO similarly activates several biological functions as a secondary messenger or chemokine, like NO [125]. HNO possesses several beneficial effects in cardiovascular physiology, such as the inhibition of platelet aggregation, vasorelaxation, and positive inotropics, with the restoration of RyR2 activity. In fact, the results of a randomized clinical trial suggested that HNO donors could be a promising treatment strategy for heart failure patients [126]. It would be beneficial to study the biological properties of HNO in detail in the future.

### 3.3. Clinical Importance of Denitrosylation

Acute or chronic inflammation accumulates nitrosative stresses in the cell and, finally, results in the cytotoxic effect of cell death. Hence, it is a useful strategy for minimizing cellular and organ damage to resolve nitrosation in the cell.

#### 3.3.1. Therapeutic Potential of Detritylation: Nitric Oxide Synthase Inhibitor

Nitrosative stresses initially originate from NO production through the subtype of NOS. For this reason, NOSis are commonly tested, especially to specify the role of NOS in disease progression [32]. NOSis are widely utilized in experimental models in a transient manner and have provided evidence for why we must administer NOSi for the treatment of disease without obvious side effects; however, we must consider that NOSis would be prescribed for a longer period than that for an animal model, which increases the risk of side effects from the cumulated ingestion of NOSi. Experimentally, there is no NOS subtype-specific inhibitor, but there is a selective inhibitor over other subtypes. In association with chronic inflammation, aberrant activation of iNOS is closely linked to disease progression and iNOS null allows for resistance against the deterioration of diseases. Moreover, 1400 W dihydrochloride is one of most iNOS-selective inhibitors that shows K_d_ = 7 nM for iNOS, and 2 μM and 50 μM for nNOS and eNOS, respectively [127]. However, the administration of 1400 W dihydrochloride tends to increase the vasoconstriction response, which emphasizes that the eNOS function is also affected by 1400 W dihydrochloride. eNOS inhibition, and thereby general hypertension, is a noteworthy side effect of NOSi usage. Actually, L-NAME is widely used to induce hypertension in rodents [103,128]. For this reason, direct inhibition with NOSi is an attractive strategy to improve nitrosative stress, but denitrosylation would be more beneficial.

#### 3.3.2. Therapeutic Potential of Denitrosylation: Complement of Glutathione

GSH mainly serves as denitrosylase in cells through transnitrosylation. In the presence of NADH or NADPH, GSNOR and GR sequentially restore GSH [14]. TRX is also involved in reducing protein denitrosylation, and TrxR again reduces TRX back in the presence of NADPH. Hence, direct supplementation of GSH from the exogenous formula, direct GSNOR/GR/TRX/TrxR inducer, indirect stimulants for GSNOR/GR/TRX/TrxR, and the restoration of NADH/NADPH would be a considerable modality for denitrosylation. GSH, a three amino acid molecule, is already approved and can be consumed as a daily nutritional supplement.

#### 3.3.3. Therapeutic Potential of Denitrosylation: Thioredoxin and Thioredoxin Reductase Inducer

Unfortunately, no chemical that directly induces or activates GSNOR/GR has been reported. Several chemicals and hormones are known to induce TRX in the limited experimental condition. Retinol [129]; estradiol, but not testosterone [130]; prostaglandin E1 [131]; and geranylgeranylacetone [132] show notable increases in TRX expression.

#### 3.3.4. Indirect Denitrosylase

##### Nicotinamide Adenine Dinucleotide and Nicotinamide Adenine Dinucleotide Phosphate

NAD(P)H is involved in diverse steps for reducing nitrosative stress as a cofactor. It is important to supply a sufficient amount of NAD(P)H [133]. NAM, also called niacinamide, is a form of vitamin B3 that is prescribed for nutritional supplementation. NAM is used for making NADH and NADPH after absorption, which are crucial cofactors [134]. Most multivitamin complexes carrying vitamin B complex generally contain high amounts of NAM. Due to its water-soluble property, NAM is almost absent of side effects, even with high consumption. NAM must be tested for a novel denitrosylation modality.

##### Antioxidants

As discussed, RNS is generally present alongside ROS. Hence, many antioxidants can be considered for novel denitrosylase. Superoxide dismutases rapidly catalyze the superoxide radical (O_2_·^−^) into O_2_ and H_2_O_2_. Because O_2_·^−^ forms ONOO^−^ with NO, superoxide dismutase successfully suppresses the generation of ONOO^−^ and subsequent tyrosine nitration. Superoxide dismutase is a notable target for reducing nitrosative stress, especially ONOO^−^ modification [135]. Although the authors did not test the S-nitrosylation profiles in their HFpEF model and after changes in the administration of NAM, alleviation of S-nitrosylation possibly contributed towards HFpEF improvement [136].

##### Nuclear Factor Erythroid 2-Related Factor 2 and Its Inducers

NRF2 is a well-established master regulator that orchestrates antioxidation, detoxification, and cell survival [137,138]. In the quiescent state, NRF2 is localized in the cytoplasm by forming a heterodimer with Kelch-like ECH-associated protein 1 (Keap1). Keap1 releases NRF2 in response to several stimuli, including high levels of ROS generation. NRF2 shuttles into the nucleus and activates its target gene transcription, such as NAD(P)H dehydrogenase (quinone) 1, glucose-6-phosphate dehydrogenase, GR, TrxR, and SOD. Collectively, NRF2 activation directly increases reductase enzyme levels (GR, TrxR, and SOD) and indirectly restores NADPH, which, in turn, strongly induces denitrosylation. In fact, we delivered NRF2 in the primarily cultured cardiac myocytes or in the heart tissue through the use of a virus and observed significant denitrosylation in the disease entity. Similarly, the direct or indirect NRF2 inducer is a promising candidate for denitrosylation. DMF is approved for relapsing-remitting MS. No direct evidence is available; however, DMF induces NRF2 expression and its target genes. Among the various effects, DMF is regarded as an antioxidant agent via NRF2 pathway activation [139]. Our group also tested the denitrosylation ability of DMF in our HFpEF experimental model [78]. Daily ingestion of DMF successfully induced NRF2 in the heart, and confirmed that overall, S-nitrosylation decreased by DMF. Although we could not elucidate whether the denitrosylation property totally depends on NRF2, we suggest that denitrosylation by DMF significantly improves HFpEF progression, at least according to our experimental model. Various chemicals known to induce NRF2 must be examined for their ability to elicit denitrosylation and ameliorate disease progression, as shown in our results. A schematic demonstration is available in Figure 2.

## 4. Conclusions and Future Perspective

In this review, we discussed how nitrosative stresses were generated in disease entities and how we can approach the neutralization of nitrosation in cells. NOS generally produces NO in physiologic and pathologic situations. In association with an excessive oxidation environment, NO forms ONOO^−^ and then induces apoptosis and irreversible tyrosine nitration. On the other hand, NO is also involved in temporal cysteine modification through S-nitrosylation. S-nitrosylation mostly regulates function complex formation and signal propagation. S-nitrosylation is an important system for regulating the physiologic activity in the cell, which is tightly controlled in the posttranslational modification base. In a disease situation, the total amounts of NO and its derivatives are generally increased, and the total S-nitrosylation/tyrosine nitration are increased accordingly. Many studies have suggested the responsible S-nitrosylation protein associated with disease development; however, it remains unclear why the target molecules are specifically modified in the disease state. In other words, NOS also exists in the physical status and produces NO to perform its physiological response, but NOS also generates cytotoxic NO modification in the pathologic status and contributes towards disease progression. Although the total expression level of NOS and the subsequent NO amount are commonly increased, whether the total amount of NO matters in disease progression or whether the abnormal coupling with a disease progression partner results in the deterioration of disease remains to be elucidated.

There are several considerable therapeutic strategies to reduce nitrosative stresses in cells. Because NOS produces NO, NOSi is useful for reducing the total NO amount in the cell. NO is important for physiologic responses, especially in vascular homeostasis and neuronal function, where disease-specific NOSi would be necessary. However, no selective or subtype-specific NOSis have been approved. NOSi is quite an attractive target molecule to develop modality; it is essential to escape or at least minimize the beneficial NO effect in human physiology.

For this reason, a denitrosylation strategy is a more notable target. The GSH/GSNO chain reaction to reduce S-nitrothiol and TRX/TrxR participates in intracellular denitrosylation processes with NADH or NADPH consumption. Hence, the restoration of a non-reducing material, such as GSH and NAD(P)H, is noteworthy to activate the denitrosylation process. Furthermore, such materials are already approved and utilized as daily supplements. It is necessary to design clinical trials testing GSH and NAD(P)H supplements in nitrosative stress dominant human disease as an adjuvant with conventional regimens. Although no direct/indirect inducers for GSNOR/GR have been reported, several possible modalities have been reported to activate TRX/TrxR. In particular, the TRX/TrxR system resolves the SS bridge to remove aggregation in the damaged cell. This possibility must be extensively studied.

As a master regulator for the antioxidant factory, NRF2 is well-established. NRF2 is generally targeted by KEAP1 and undergoes polyubiquitination-dependent clearance. KEAP1 releases NRF2 in response to accumulated oxidative stress. NRF2 shuttles into the nucleus and induces the transcription of its target genes. There are numerous target genes, direct or indirect, and these genes, in turn, participate in various steps in the elimination of oxidative stress. Although NRF2 targets the antioxidation process, many of its target genes also reverse nitrosative stress. Furthermore, several SODs induced by NRF2 can suppress the generation of ONOO^−^, which can reduce RNS-induced cellular damage and cytotoxicity. Several NRF2 inducers, in direct and indirect, ways are available (summarized in Table 1). Still, limited evidence is reported to prove its denitrosylation property, but it has the potential for a future strategy for nitrosative stress diseases. Unfortunately, NRF2 is a double-edged sword. NRF2 is associated with cancer cell survival due to its powerful antioxidant effect [140,141]. NRF2 strongly suppresses oxidative stresses and thereby apoptotic cell death. High levels of NRF2 in solid tumors are strongly correlated with malignancy and a poor prognosis. However, NRF2-mediated antioxidation and the cytoprotection effect are attractive targets for chronic inflammation diseases.

Fortunately, NRF2 did not accelerate the transformation of normal cells, and KEAP1 successfully controlled the NRF2 activity. To bypass possible cancer risks from NRF2, more specific approaches, such as those that utilize the direct activator of GSNOR/GR for denitrosylation, will be necessary in the future, rather than general NRF2 inducer-based approaches.

## Figures and Tables

**Figure 1 ijms-22-09794-f001:**
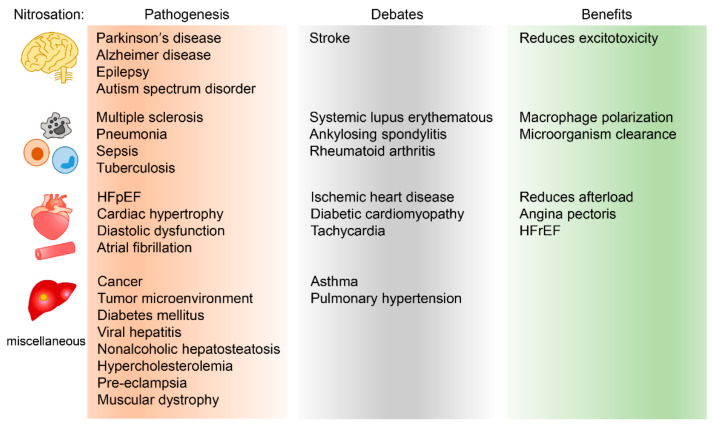
Clinical implications of nitrosation in human disease. HFpEF, heart failure with preserved ejection fraction; HFrEF, heart failure with reduced ejection fraction.

**Figure 2 ijms-22-09794-f002:**
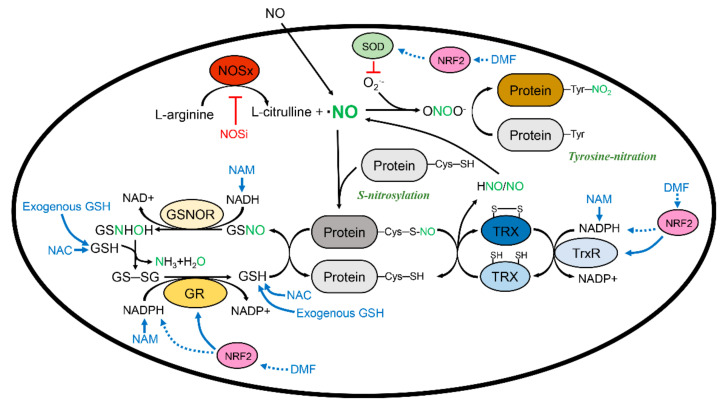
Schematic demonstration of the nitrosation and endogenous reducing system. Blue arrow: direct activation; dashed arrow: indirect activation; red blunted line: inhibition. Cys, cysteine; DMF, dimethyl fumarate; GR, glutathione reductase; GSH (γ-l-glutamyl-l-cysteinyl-glycine), glutathione; GSNHOH, N-hydroxysulfinamide; GSNO, S-nitrosoglutathione; GSNOR, S-nitrosoglutathione reductase; GSSG, glutathione disulfide; GSSOH, glutathione sulfinic acid; HNO, nitroxyl; NAC, N-acetyl-L-cysteine; NADH, nicotinamide adenine phosphate; NADPH, nicotinamide adenine dinucleotide phosphate; NAM, nicotinamide; NO, nitric oxide; NOS, nitric oxide synthase; NOSi, NOS inhibitor; NRF2, nuclear factor erythroid 2-related factor 2; ROS, reactive oxygen species; SOD, superoxide dismutase; TRX, thioredoxin; TrxR, thioredoxin reductase; Tyr, tyrosine.

**Table 1 ijms-22-09794-t001:** Experimental molecules for reducing nitrosation.

Category	Mechanism	Reference
Reduces NO Production		
NOS inhibitor		
Proadifen hydrochloride	nNOS/AchR inhibitor	[142]
N(ω)-propyl-L-arginine	nNOS selective inhibitor	[85]
S-methyl-L-thiocitrulline	nNOS selective inhibitor	[85]
7-nitroindazole	nNOS selective inhibitor	[40]
1400 W	iNOS selective inhibitor	[98]
(S)-methylisothiourea sulfate	iNOS selective inhibitor	[143]
N6-(1-iminoethyl)-l-lysine dihydrochloride	iNOS selective inhibitor	[59]
L-N5-(1-iminoethyl)ornithine	Nonselective NOS inhibitor	[144]
N(G)-monomethyl-L-arginine acetate	Nonselective NOS inhibitor	[92]
N-nitro-L-arginine methyl ester	Nonselective NOS inhibitor	[103]
N-nitro-L-arginine	Nonselective NOS inhibitor	[145]
2-(4-carboxyphenyl)-4,4,5,5-tetramethylimidazoline-1-oxy-3-oxide	NO scavenger	[146]
**Induces denitrosylation**		
GSH/GSNO system		
Dietary GSH	GSH supplement	[147]
N-acetyl-L-cysteine	GSH precursor	[79]
Nrf2	GR inducer	[148]
TRX/TrxR system		
Retinol	TRX inducer	[129]
Estradiol	TRX inducer	[130]
Prostaglandin E1	TRX inducer	[131]
Geranylgeranylacetone	TRX inducer	[132]
Nrf2	TrxR inducer	[149]
Indirectly involved in denitrosation		
Superoxide dismutase	ROS reducer	[92]
Niacin (Vitamin B_3_)	NAD(P)H precursor	[134]
Nicotinamide	NAD(P)H precursor	[134]
Dimethyl fumarate	Nrf2 inducer	[78]
Oltipraz	Nrf2 inducer	[150]
Sulforaphane	Nrf2 inducer	[151]
Bardoxolone-methyl	Nrf2 inducer	[152]

AchR, acetylcholine receptor; GR, glutathione reductase; GSH, glutathione; GSNO, S-nitrosoglutathione; iNOS, inducible nitric oxide synthase; NAD(P)H, nicotinamide adenine dinucleotide (phosphate); nNOS, neuronal nitric oxide synthase; NO, nitric oxide; NOS, nitric oxide synthase; Nrf2, nuclear factor erythroid 2-related factor 2; ROS, reactive oxygen species; TRX, thioredoxin; TrxR, thioredoxin reductase.

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
