# Peer review of "Nitrosative Stress and Human Disease: Therapeutic Potential of Denitrosylation"

_ijms, 2021, doi:10.3390/ijms22189794_

Round 1

Reviewer 1 Report

The present paper originally and interestingly reports the therapeutic potential of denitrosylation in nitrosative stress and human disease. Although many different molecular  mechanisms were detailed with numerous partners opening different opportunities to regulate nitrosative stress, it is surprising that regulations of the” second messenger” cyclic nucleotides which are directly modulated by NO were not discussed in the present report although they govern inflammatory responses at the level of  many organs, tissues, cells and subcellular compartments. Therefore, keeping in mind that cyclic nucleotide phosphodiesterases (PDEs) by controlling cAMP and cGMP levels might control  protein phosphorylations as well as protein expressions, cell cycle, it is necessary to include and discuss in the manuscript that PDEs being regulated by NO mainly participate in human disease.

Page 2, Line 87: a ref explaining the molecular mechanism of relaxation induced by NO in smooth muscle is missing.

Page 3: line 101 to 110: it is necessary to report that beneath cGMP formation, cGMP level is mainly dependent of  the PDE super family which rapidly degrade cyclic nucleotide and control normal and physio-pathological intracellular signaling.

Fig 1 reports pathogenesis in which nitrosation might be implicated. Interestingly, these pathogeneses are also dependent on PDE dysregulations  in numerous cell compartments (Br J Pharmacol. 2012 ;165(5):1288-305) as well as for those which are under debates, opening new research possibilities.

Page 5,  In accordance with authors it is of interest to note that PDEs have been also characterized  in different subcellular structures, such as cytoplasm, nucleus, mitochondria  and has been implicated in neurodegenerative stress: Alzheimer’s  and Parkinson’s diseases, and epilepsy.

 Page 7, line 322, “Similarly” does not correspond to the sentence meaning; “in opposite” might be more appropriated.

Page 8, “Though the authors did not clarify the relevant target.. both IkB and NFkB..” However, to go further it should be of interest to know that PDE4 inhibitor overcomes inflammation related to both IkB and NFkB as well as LPS-induced effects.

Page 10 :”An experimental sepsis…eNOS activation through pro-inflammatory cytokine such as TNFα, IL-1, Il-6..”. In that field it must be pointed out that these cytokines are  initially regulated by PDE4 that specifically hydrolyzed cAMP (see Can. J. Physiol. Pharmacol. 91: 353–361 (2013) ; PLoS One 2012;7(1):e28899. doi: 10.1371). Furthermore, it is well established that NO potentializes vasorelaxation induced by  specific PDE inhibitors, and in that way might participate to hypotensive crisis.

Page 11, line 500; it is reported that “female tends to express more abundant levels of eNOS…/males in I/R injury “ What will happen when they are smoking?

Lines 519-520: it is necessary to give a reference reporting I the cardiovascular system dual effect of NO on cGMP-cAMP cross-regulation , as much as  this was shown to be  mediated by PDEs.

Page 18, Authors pointed out the interest to investigate the molecular mechanism of the beneficial effect of HON  in the cardiovascular system. It might be necessary to give the reference Circ Heart Fail. 2013 Nov; 6(6): 1250–1258.  showing  that this is effective in the human. Is it some hypothesis on the molecular mechanism?

Reviewer 2 Report

Dear Authors,

Although I have suggested publishing the article in its current form, I would like to point out two minor errors:

  1. At the end of line 18, the full stop is missing from the end of the sentence (before "two").
  2. In line 950 the necessity of disease specific NOSi is mentioned (which currently does not exist). So in the next sentence, "no non- selective or subtype-specific NOSis..." should be replaced by "no selective or subtype-specific NOSis....". In its current form, it is a bit confusing.

In addition, I find this Review to be excellent, logical, and inspiring for the future direction of research.

Round 2

Reviewer 1 Report

The authors have partially answered to referee questions and suggestions by giving a global answer only focused on NO and cGMP, although a cross-talk between cAMP and cGMP mediated by PDEs governs physio-pathological responses has been clearly established.

Since ref 19 is focused on erectile dysfunction another ref more appropriated would be given.

Author Response

Reviewer 1

We thank Reviewer 1 for the constructive comments on our manuscript.

  1. cGMP-cAMP cross-talk

The authors have partially answered to referee questions and suggestions by giving a global answer only focused on NO and cGMP, although a cross-talk between cAMP and cGMP mediated by PDEs governs physio-pathological responses has been clearly established.

Response: We have discussed cGMP-cAMP cross talk according to reviewer’s suggestion.

  1. Reference citation

Since ref 19 is focused on erectile dysfunction another ref more appropriated would be given.

Response: We understand the concern of the reviewer. In this regard, we have replaced reference 19 with the reference regarding pulmonary hypertension (Archer, S.L.; Michelakis, E.D. Phosphodiesterase type 5 inhibitors for pulmonary arterial hypertension. N Engl J Med 2009, 361, 1864-1871, doi:10.1056/NEJMct0904473.)